# Creation of a shortened version of the Sleep Disorders Questionnaire (SDQ)

**Kathleen Biard[1]☯, Joseph De Koninck[1,2]☯, Alan B. Douglass[2,3]☯ \***

**1** School of Psychology, Faculty of Social Sciences, University of Ottawa, Ottawa, Ontario, Canada,
**2** University of Ottawa Institute of Mental Health Research at the Royal Ottawa Mental Health Center,
Ottawa, Ontario, Canada, **3** Psychiatry Department, Faculty of Medicine, University of Ottawa, Ottawa,
Ontario, Canada

☯ These authors contributed equally to this work.
\* alan.douglass@theroyal.ca

pone.0288216

University of Medical Sciences, ISLAMIC
REPUBLIC OF IRAN

**Data Availability Statement:** All relevant raw data
are within the manuscript and its Supporting
Information files. Full text of the SDQ-2

## Abstract

The 176-item Sleep Disorders Questionnaire (SDQ) was initially developed using canonical discriminant function analysis on 4 groups of sleep disorder patients, but it was never studied by factor analysis in its entirety. Several authors have criticized 2 of its subscales as being confounded with each other, and its narcolepsy scale as substantially over-diagnosing narcolepsy. This study describes its first exploratory factor analysis (EFA), the intent of which was to reassess item membership on the 4 existing subscales and to derive new scales to improve differential diagnosis between patient groups. It was also hoped that EFA could reduce the total number of questions, to increase speed of completion. The EFA was performed on the anonymized SDQ results from a retrospective review of the charts of 2131 persons from 7 sleep disorders clinics and research centers. Factors were assessed via scree plots and eigenvalues. The EFA identified four main factors: insomnia, daytime sleepiness, substance use, and sleep-disordered breathing. The insomnia factor had 3 subfactors: psychological symptoms of insomnia, subjective description of insomnia, and insomnia due to periodic limb movements. The sleepiness factor had two subfactors: daytime sleepiness and neurological symptoms of narcolepsy. The novel substance use factor was homogeneous, as was the sleep-disordered breathing factor. Importantly, the EFA reassigned items from the original SDQ's NAR, PSY, and PLM subscales to five of the new subscales. The Sleep Apnea (SA) subscale emerged mostly unchanged. The 7 resulting factors comprised only 66 items of the original 176-item SDQ. These results have allowed the creation of a new shorter questionnaire, to be called the SDQ-2. External validation of the SDQ-2 is currently underway. It will likely prove to be a superior differential diagnostic instrument for sleep disorders clinics, compared to the original SDQ.

## Introduction

The aim of the original Sleep Disorders Questionnaire (SDQ) was to provide a self-report tool to assess the risk of having one of several sleep disorders. It was developed between 1983 and

questionnaire and its scoring sheet are also uploaded.

**Funding:** KB received a Canada Graduate Scholarship (CGS) from the (Canadian) Natural Sciences and Engineering Research Council (NSERC) that supported her PhD studies on this project. https://www.nserc-crsng.gc.ca/ The funder had no role in study design, data collection and analysis, decision to publish, or preparation of the manuscript.

**Competing interests:** I have read the journal's policy and the authors of this manuscript have the following competing interests: Drs. Biard and De Koninck declare no competing interests. Dr. Douglass claims copyright of the resulting SDQ-2 questionnaire and may profit from its future commercialization. He is also a paid consultant to the Ottawa Police Association, which is sponsoring a validation study of the SDQ-2. This does not alter our adherence to PLOS ONE policies on sharing data and materials.

**Abbreviations:** Berlin Questionnaire, screening questionnaire for sleep apnea; BMI, Body Mass Index (kg/m²); EFA, exploratory factor analysis; ESS, Epworth Sleepiness Scale; GAD-7, Generalized Anxiety Disorder, 7-item scale; MIES, Moral Injury Events Scale; NAR, PSY, PLM, SA, subscales of the original SDQ; OSA, obstructive sleep apnea; PCL-5, PTSD Checklist for DSM-5; PHQ-9, Patient Health Questionnaire (depression); PLM, periodic limb movements; PSQI, Pittsburgh Sleep Quality Index; RLS, Restless Legs Syndrome; ROC, receiver operating characteristics (analysis); SDB, sleep disordered breathing; SDQ, the original Sleep Disorders Questionnaire, 1994; SDQ-2, second edition of the SDQ, derived in the present paper; SQAW, Stanford University's Sleep Questionnaire and Assessment of Wakefulness; STOP-Bang, screening questionnaire for sleep apnea; TIA, transient ischemic attack; CAGE, alcohol screening questionnaire; ISI, Insomnia Severity Index.

1994 by Douglass and colleagues [1] at Stanford University from their existing Sleep Questionnaire and Assessment of Wakefulness (SQAW) [2]. The latter had been difficult to use clinically due to its great length, over 800 items, and its many different response sets. SDQ retained only 175 of its items, which were then grouped, simplified, re-worded, and all changed to 5-point Likert scales. Body mass index (BMI) was added as the 176th item, to be entered by a clinician.

SDQ items were deemed to have adequate face validity when assessed by three Stanford sleep specialists, and were validated against polysomnographic diagnoses in a canonical discriminant function analysis of 519 patients [1]. This resulted in four subscales characteristic of patients with common sleep disorders (subscale name in brackets): narcolepsy (NAR), sleep apnea (SA), periodic limb movement disorder (PLM) and sleep disturbance due to psychiatric illness (PSY). Receiver Operating Characteristics (ROC) analysis for each subscale demonstrated adequate sensitivity, specificity, and positive / negative predictive values. Item test-retest reliability of the SDQ items was assessed by Pearson correlation over an interval of 3 to 4 months in 130 patients and 71 normal controls [3]; the range of correlation was from $r = 0.308$ to 0.985, mean $r^2 = 0.636$.

With its focus on diagnosis, SDQ differed from scales like the Pittsburgh Sleep Quality Index (PSQI) [4], which focus on subjective descriptions of sleep that correlate most strongly with psychological mood and anxiety questionnaires. Another focus of SDQ has been the identification of sleep disorders that are co-morbid with other medical and psychiatric diagnoses. This was exemplified by a study [5] using SDQ and the NoSAS questionnaire to look for sleep apnea in patients with diagnosed depression. Similarly, a controlled study [6] showed significant elevations of the PSY and PLM scales in patients with COPD.

Using the German translation of the SDQ, a study [7] of 3521 patients undergoing polysomnography showed significant sex differences by ANOVA within the 4 SDQ scales, which the authors postulated might explain the under-recognition of sleep-disordered breathing (SDB) in women.

A brief review of the literature regarding the 4 existing subscales of the SDQ and problems with their usage is as follows:

## SDQ-PLM subscale

A 2006 study of 4,901 Swiss pharmacy customers by Schwegler [8] factor analyzed 45 items from the French and German translations of the SDQ. Their goal was to confirm the uniqueness of the 4 original SDQ subscales and to use the Epworth Sleepiness Scale [9] as an external validator. While their Cronbach's alpha coefficients for subscale homogeneity were similar to those in the original SDQ paper, they found that 60% of respondents scored above the originally-proposed pathological cut-off score of the PLM subscale (which they called "RLS scale"). They also found that the SDQ-PSY subscale had Spearman correlations of nearly 0.50 with both the PLM and NAR subscales, belying its intended purpose of identifying sleep disturbance in patients with psychiatric illness. To date, few other papers have reported use of the SDQ-PLM subscale. A validation by Valencia-Flores [10] found a correlation (Spearman's rho = 0.59) between the SDQ-PLM subscale and the PLM index on polysomnography in 14 patients with systemic lupus erythematosus. This subscale has also been used to screen for causes of sleep problems in dementia patients [11]. Overall, SDQ's PLM subscale appeared to be tapping into more general features of insomnia rather than those due only to true RLS/ PLMD.

## SDQ-NAR subscale

This subscale was intended to be sensitive to the symptoms of patients suffering narcolepsy-cataplexy (narcolepsy type 1), but it also proved to include items that were endorsed by respondents who were sleepy due to other causes. Another surprise in the Schwegler study [8] was that 10% of their respondents scored above the 90th percentile on the NAR subscale. Since the population prevalence [12] of narcolepsy is less than 50/100,000, the SDQ-NAR subscale obviously grossly over-diagnosed narcolepsy. These results suggested that the NAR subscale might be reflecting non-narcoleptic sleepiness in the general population, in addition to neurological symptoms of narcolepsy. Unexpectedly, Parkinson Disease patients also scored significantly higher on the NAR subscale than controls [13], although this was attributed to side-effects of their dopaminergic medications rather than to narcolepsy.

## SDQ-SA subscale

Some papers refer to this as "SA-SDQ", but the preferred acronym is "SDQ-SA". Its diagnostic intent was similar to that of other obstructive sleep apnea (OSA) questionnaires in the literature, such as the Berlin [14], STOP-Bang [15], NoSAS [16], and the OSA50 [17]. In contrast to these, however, the SDQ-SA did not require any physical measurements or interaction with a clinician. SA is the SDQ subscale most commonly reported in the literature [18–21], likely due to its high sensitivity, specificity, PPV, and NPV when compared [22] to other sleep apnea questionnaires. Recent meta-analyses and reviews [23–26] have also supported the diagnostic usefulness of the SA subscale.

A much larger Swiss pharmacy study [27] of nearly 200,000 persons employed the SDQ-SA (called "SAS" by the authors) along with the Epworth Sleepiness Scale (ESS) to predict motor vehicle accidents due to sleepiness. They found that SDQ-SA subscale homogeneity by Cronbach's alpha was again similar to the value in the original Douglass paper at rho = 0.74.

Malow [28] found that SDQ-SA scores were significant predictors of ESS scores in 158 epilepsy patients, whereas anti-epileptic medication dose or seizure frequency were not. Weatherwax [29] suggested that the original cutoff scores for diagnosing OSA in the general population should be lowered for epilepsy patients. The SDQ-SA subscale has also been used to identify OSA in patients with transient ischemic attacks (TIAs) [30–33] and in those with morbid obesity [34].

## SDQ-PSY subscale

Scores on this subscale are high in hospitalized patients with major mood disorders or psychosis, likely due to the insomnia and sleep fragmentation often reported by these patients. It was partially validated [35] by correlation with the Carroll Depression Scale [36] in 44 patients diagnosed with Major Depressive Disorder (r = 0.70, p < 0.0001). It also showed significant negative correlation with sleep efficiency on the polysomnograms of these same patients (r = -0.30, p < 0.05). Brower [37] used this subscale to assess the subjective sleep complaints of 74 patients recovering from alcohol dependence in a search for symptoms predictive of alcoholism relapse. Relapsers proved to have higher baseline scores on the SDQ-PSY subscale than those who abstained over a follow-up period of 3–12 months. Some authors, including Brower, have however argued that the SDQ-PSY subscale is actually an insomnia subscale, based on their own studies [38, 39]. In support, Sweere [40] performed a cluster analysis on a Dutch translation of 89 items from the SDQ and found a clear insomnia subscale in addition to 2 others that closely resembled the SDQ's SA and NAR subscales.

Taken together, the above findings suggest that the use of SDQ in non-clinical populations sometimes substantially over-diagnoses sleep pathologies. They also imply that some SDQ

subscales are confounded with each other, possibly due to items concerning insomnia appearing on more than one subscale. Some of these problems could also be due to the fact that the original SDQ scales were derived not by factor analysis but rather by canonical discriminant function analysis on responses from 4 polysomnographically-diagnosed groups of sleep disorder patients.

Over the past 35 years, SDQ has been administered to tens of thousands of persons in medical and surgical outpatient clinics, general hospital inpatients, psychiatric hospitals, sleep disorders centers, and university psychology departments. It has been translated into French, Dutch, German, Italian, Spanish, Greek, Arabic, and Turkish; therefore, there is a substantial pool of data available for secondary analysis if the above problems could be resolved.

The rationale for the present study was to perform secondary analysis of new data to reassess the factor structure of the 4 existing subscales via EFA. It was also hoped that new subscales could be derived, such as a specific insomnia subscale. Finally, it was hoped to reduce the total number of questions from 176 to a more manageable size for the SDQ-2 revision.

## Methods

In all, SDQ responses of 2185 adults (1543 males, 642 females) were obtained for secondary analysis from a variety of hospitals and universities at which the SDQ had been completed prior to a laboratory nocturnal polysomnogram Responses were gathered over the period 1986–2007, data collation was completed in 2008, and computer analysis took place from 2008–2011. These data therefore represented a sample of convenience and not a population sample planned in advance. Since most of these sites were either psychiatric facilities or general sleep disorders centers, the focus of the SDQ is necessarily slanted towards symptoms experienced by such patients rather than the general population. Polysomnographic data are not reported here but will be reported in future papers on subscale validation.

Of the above total, 1356 participants were patients of Dr. Douglass at various hospitals, meaning that he was aware of their identities. The IRB allowed SDQs on these charts to be retrospectively reviewed with the proviso that their anonymity be ensured by assigning code numbers as case identifiers, instead of patient names or hospital chart numbers, before being included in the analysis. In the case of patients from other doctors or hospitals, code numbers were used exclusively. The data from 54 cases (2.5% of total) were discarded because more than 20% of items were unanswered, leaving the data of 2131 persons (1497 males, 634 females) for the EFA. This gave a patient-to-questionnaire-item ratio of 12:1, which is commonly regarded as adequate for multivariate analysis. Any remaining missing values among the 176 items were replaced by using a multiple imputation procedure ("PROC-MI", SAS version 9.1, SAS Institute, Cary, North Carolina), which operates via an iterative process. This version of SAS software was also used to compute the EFAs ("PROC-FACTOR") using a varimax factor rotation of both orthogonal and oblique types. Eigenvalues and scree plots were used to determine the final number of factors to be retained.

We then examined the items in each factor by condition index to determine if there was significant collinearity; if so, all but one of the collinear variables was deleted. After the main factors had been determined, a secondary factor analysis was run on some factors that proved to be heterogeneous, in order to identify possible subfactors.

The original SDQ had 9 gender-specific items (4 for males, 5 for females). We dealt with these by removing all of them from the main factor analysis. We then split the file by sex and analyzed the responses of males and females separately, after removing items from the opposite sex. This also allowed us to verify whether the factor structure was similar in men and women.

More detailed explanations of the statistical methods and results are shown in the Supporting Information section.

Under an authorization dated 2008-04-11 by the *University of Ottawa Social Sciences and Humanities Research Ethics Board* (File 02-08-16), secondary use of anonymized data from the institutions listed in the attached "Thank-you Attestation" was approved. The project was also approved in 2008 (File number 2008001) by the *Research Ethics Board of the Royal Ottawa Hospital.*

## Results

Clinical and demographic details about the SDQ respondents are found in Table 1. Fifty-four cases were deleted due to missing data, giving a case-wise missing rate of 2.5%. After deletions, the remaining variable-wise average missing rate was 3.1%, all of which were replaced by imputed values.

The EFA revealed four Main Factors when using a visually-determined scree cutoff, all of which had an eigenvalue > 5.0 (Fig 1).

The percentage of overall variance explained by these factors is shown in Table 2.

Inter-correlations of the Main Factors' oblique rotations are shown in Table 3.

Secondary EFA of Main Factor 1 resulted in 3 subfactors with clinical relevance, while the same process on Main Factor 2 resulted in 2 subfactors. Therefore, 7 useful factors were extracted from the SDQ data. Scree plots of the secondary EFAs for Main Factors 1 to 4 are shown in S1 Fig, parts "A" to "D".

Item loadings on the Main Factors and their subfactors are shown in Tables 4–7. In these tables, items that were also present on the four original SDQ subscales are identified with different markers. Items selected for the final SDQ-2 subscales had to have item loadings of 0.4 or greater on the above factors.

Subscale homogeneity was calculated via Cronbach's alpha for each of the 7 subscales and is shown below.

Collinearity diagnostics on all factors are shown in S1 Appendix, along with the names of items that were removed to resolve the collinearity. The number of items removed from Main Factors 1–4 by this process were 0, 2, 1, and 2, respectively.

In addition to the four Main Factors, another 15 factors resulted from the primary EFA, numbered 5–19, all with eigenvalues > 1. The subject matter of these factors and their item

**Table 1. Characteristics of the respondents (n = 2185[c]).**

| | | MALES | | | | FEMALES | |
|---|---|---|---|---|---|---|---|
| **DIAGNOSTIC GROUP** | **N** | **Mean Age** | **SD** | | **N** | **Mean Age** | **SD** |
| Psychiatric Patients[a] | 574 | 45.18 | 13.41 | | 296 | 41.85 | 12.67 |
| General Practice Referrals[b] | 899 | 45.23 | 13.77 | | 338 | 44.78 | 13.79 |
| Normal control subjects | 70 | 38.70 | 12.71 | | 8 | 38.61 | 12.73 |
| **SUBTOTALS** | **1543** | **44.92** | **13.59** | | **642** | **43.35** | **13.27** |
| **Male + Female TOTAL** | | | | **2185[c]** | | | |
| **54 removed (missing data)** | **1497** | | | **2131** | **634** | | |

[a] Hospital inpatients and outpatients with mood disorders, schizophrenia, anxiety disorders, PTSD, or alcoholism. They were usually studied clinically to rule-out a possible physical sleep disorder but some took part in research sleep studies (baseline nights). All patients were studied without CPAP or other physical treatment for sleep apnea.

[b] Patients with a medical / surgical diagnosis who were referred by their general practitioner or specialist to rule out sleep apnea or physical causes of insomnia.

[c] A total of 54 respondents (2.5% of total) were removed due to having >20% missing items.

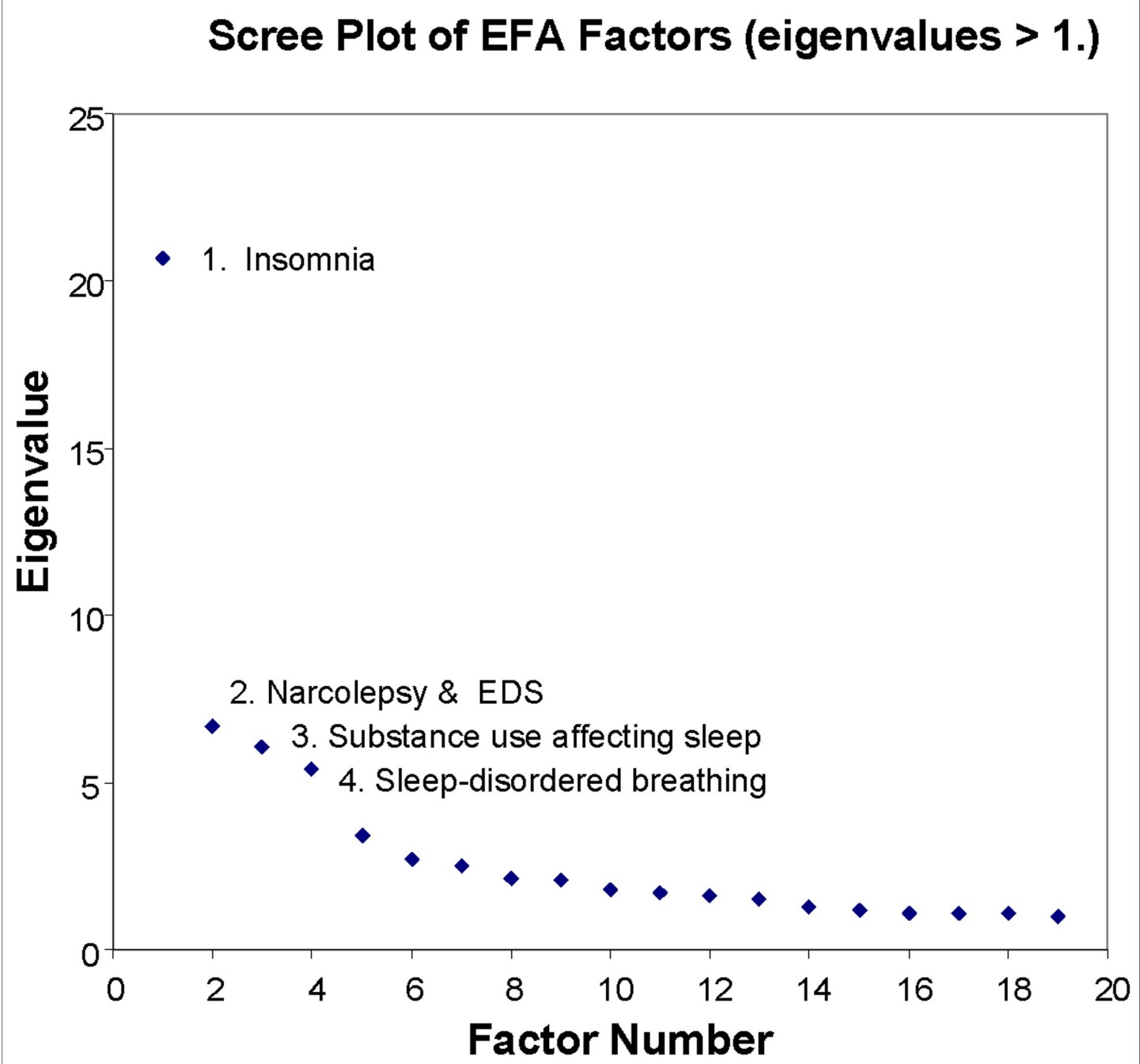

**Fig 1. Scree plot of all factors extracted by the EFA.** The first 4 factors all have eigenvalues greater than 5, and are referred to as Main Factors in the text.

loadings are summarized in S1 Table. While they do not uniquely identify any clinical sleep disorders, they may be of interest to some readers considering the large pool of respondents in this study. Examples of their content include: psychosomatic causes of insomnia; seizures and parasomnias; tonsil and adenoid problems; and sleepiness due to shift work.

Below is a detailed description of the 7 factors identified above and their homogeneity (Cronbach's alphas). These factors are proposed to be the subscales of the new SDQ-2.

**Table 2. Percentage of overall variance explained by the 4 main factors.**

| Factor[a] | Orthogonal Rotation | Oblique Rotation | |
|---|---|---|---|
| | | Shared variance | Unique variance |
| 1 | 12.76 | 15.43 | 8.92 |
| 2 | 8.89 | 11.53 | 6.69 |
| 3 | 6.68 | 8.67 | 5.91 |
| 4 | 6.57 | 6.46 | 6.43 |
| **TOTAL** | **34.90** | **42.09** | **42.09** |

[a] Factor numbers in the table refer to those in Fig 1.

## Main factor 1: Insomnia

Subfactor 1A "InsPsy": Psychological features of insomnia ($\alpha = 0.919$)

Subfactor 1B "InsSubj": Subjective description of insomnia severity ($\alpha = 0.831$)

Subfactor 1C "InsPLM": Insomnia due to leg sensations ($\alpha = 0.823$)

Table 4 lists items that load on Main Factor 1 using an orthogonal rotation. Results for the oblique rotation were similar. This factor incorporates 5 items from the original SDQ-PSY subscale (marked with "Ψ"), as well as 5 from the original SDQ-PLM subscale (marked with "L"). Also shown are 7 of the SDQ items from Brower's 1998 insomnia subscale [37]. The scree plot of the secondary factor analysis of Main Factor 1 (S1 Fig, Part A) does not have a clear visual cut-off but there are 3 factors having an eigenvalue > 1. When using an orthogonal rotation, Subfactor 1A explained 6.1% of the overall dataset variance, subfactor 1B explained 5.6% and subfactor 1C explained 2.8%. Subfactor 1A included psychological items such as "my sleep is disturbed by worrying." Subfactor 1B is a subjective description of insomnia severity, with items such as "I get too little sleep at night." The 4 items on Subfactor 1C describe limb movements disturbing sleep.

Items 17 ("sleep is disturbed by noise") and 114 ("use prescription hypnotics to sleep") only approached criterion on subfactors A and B (loadings of 0.39 and 0.37 respectively) but will be included in those subfactors due to face validity. Items 3, 13 and 45 co-load on both subfactors A and B. Item 13 ("I fear not being able to get back to sleep after waking") will be associated with subfactor 1A while item 3 ("trouble getting to sleep at night") and item 45 ("suspect that I have insomnia") will be associated with subfactor 1B.

## Main factor 2: Narcolepsy and daytime sleepiness

Subfactor 2A "EDS": Excessive Daytime Sleepiness ($\alpha = 0.827$)

Subfactor 2B "NAR": Narcolepsy symptoms ($\alpha = 0.829$)

**Table 3. Inter-correlations of the 4 main factors (Oblique rotation).**

| | Factor1 | Factor2 | Factor3 | Factor4 |
|---|---|---|---|---|
| Factor1 | 1 | 0.264 | 0.189 | -0.021 |
| Factor2 | | 1 | 0.097 | 0.069 |
| Factor3 | | | 1 | -0.091 |
| Factor4 | | | | 1 |

These intercorrelations demonstrate adequate independence of the Main Factors, considering that Factors 1 and 2 have subfactors. This table was calculated on Factors from which collinear items had been removed.

**Table 4. Item loadings on each subfactor of main factor 1 (orthogonal rotation).**

| Item | 1A | 1B | 1C | Source[a] | Brief Item Description |
|---|---|---|---|---|---|
| 34 | **0.801** | 0.187 | 0.173 | - | sleep disturbed by worrying |
| 8 | **0.787** | 0.121 | 0.172 | - | worry about things at bedtime |
| 33 | **0.761** | 0.169 | 0.196 | Ψ | sleep disturbed by sadness or depression |
| 7 | **0.737** | 0.116 | 0.195 | Ψ | feel sad and depressed at bedtime |
| 32 | **0.711** | 0.268 | 0.196 | - | sleep disturbed by racing thoughts |
| 6 | **0.681** | 0.201 | 0.129 | Ψ | thoughts race through my mind at bedtime |
| 10 | **0.625** | 0.398 | 0.176 | - | afraid of not being able to go to sleep |
| 36 | **0.587** | 0.370 | 0.183 | - | being unable to get back to sleep if should wake |
| 130 | **0.566** | 0.226 | 0.060 | - | mental stress, worry, or anxiety worsens sleep |
| 13 | **0.537** | **0.487** | 0.125 | - | not able to get to sleep once awake at night |
| 3 | **0.517** | **0.510** | 0.144 | Ψ* | have trouble getting to sleep at night |
| 17 | 0.390 | 0.240 | 0.138 | - | sleep is disturbed by noise |
| 2 | 0.174 | **0.802** | 0.177 | * | a poor night's sleep |
| 87 | 0.169 | **0.761** | 0.182 | * | problem with sleep |
| 14 | 0.195 | **0.731** | 0.285 | - | night sleep is restless and disturbed |
| 1 | 0.209 | **0.714** | 0.097 | * | too little sleep at night |
| 44 | 0.121 | **0.707** | 0.180 | * | feel that sleep is abnormal |
| 4 | 0.132 | **0.685** | 0.185 | L* | wake up often during the night |
| 45 | 0.432 | **0.547** | 0.088 | L* | respondent suspects they have insomnia |
| 154 | 0.295 | **0.469** | 0.022 | L | longest wake period at night |
| 43 | 0.368 | **0.403** | 0.084 | Ψ* | been unable to sleep at all for several days |
| 114 | 0.308 | **0.367** | 0.147 | - | prescription hypnotics *[kept due to face validity]* |
| 153 | -0.263 | **-0.552** | -0.040 | - | hours of sleep at night *[reverse coded on SDQ-2]* |
| 31 | 0.124 | 0.209 | **0.837** | L | sleep disturbed by "restless legs" |
| 12 | 0.151 | 0.207 | **0.808** | L | have "restless legs" when falling asleep |
| 35 | 0.323 | 0.167 | **0.678** | - | sleep disturbed by muscular tension |
| 9 | 0.370 | 0.183 | **0.632** | - | muscular tension at bedtime |

Subfactor loadings are shown in order of decreasing magnitude. Loadings in gray highlight indicate factor loadings of 0.4 or greater that were selected as SDQ-2 scale items.

**Item:** Numerical label of this item on the original SDQ.

**1A:** Subfactor related to psychological causes of insomnia.

**1B:** Subfactor related to a subjective description of insomnia severity.

**1C:** Subfactor related to insomnia attributed to leg sensations.

[a]**Source**: Indicates item appeared on a subscale of the original SDQ, as follows:

**L:** SDQ-PLM subscale.

**Ψ:** SDQ-PSY subscale.

"*": Brower's insomnia subscale [37] that predicts alcoholism relapse.

"-": Was not a member of an SDQ subscale.

Table 5 lists items that have a loading of 0.4 or higher on either the orthogonal or oblique rotations on this factor. Main Factor 2 incorporates 13 of the 15 items from the original SDQ-NAR (narcolepsy) subscale, which are marked with "#".

The scree plot (S1 Fig, part B) from the secondary factor analysis on Main Factor 2 showed a clear visual cut-off point after the second factor. Subfactor 2A explained 4.9% of the overall dataset variance and included items that describe subjective daytime sleepiness from any cause ("I am sometimes too sleepy to drive"). Subfactor 2B explained 4.6% of the overall variance and describes classical symptoms of narcolepsy ("I am paralyzed when falling asleep").

**Table 5. Item loadings on each subfactor of main factor 2 (orthogonal rotation).**

| Item | 2A | 2B | Source | Brief Item Description |
|------|------|------|--------|------------------------|
| 55 | **0.834** | 0.149 | # | very sleepy during the day, struggle to stay awake |
| 41 | **0.714** | 0.101 | - | sometimes very sleepy in the daytime, in cycles |
| 58 | **0.700** | 0.201 | # | trouble doing job because of sleepiness / fatigue |
| 59 | **0.652** | 0.176 | # | too sleepy to drive |
| 56 | **0.599** | 0.144 | # | fall asleep eating, on phone, in bus, TV, reading, etc. |
| 42 | **0.582** | 0.272 | # | have slept/been sleepy for several days at a time |
| 158 | **0.518** | 0.136 | - | number of daytime naps |
| 157 | **0.467** | 0.215 | - | car accidents due to sleepiness |
| 91 | **0.429** | 0.100 | - | sleeping more than previously |
| 115 | 0.281 | 0.280 | - | prescription drug to stay awake in day |
| 161 | -0.290 | -0.042 | - | refreshed for how long after day nap |
| 39 | 0.415 | **0.743** | # | Feel paralyzed after a nap |
| 60 | 0.126 | **0.681** | # | hallucinations before/after nap while awake |
| 11 | 0.034 | **0.661** | # | paralyzed when falling asleep |
| 40 | 0.047 | **0.657** | # | hallucinations after waking in the morning |
| 67 | 0.243 | **0.617** | # | sudden muscular weakness with strong emotions |
| 61 | 0.228 | **0.595** | - | vivid dreams during naps |
| 66 | 0.193 | **0.585** | # | "weak knees" when laughing |
| 64 | 0.313 | **0.495** | - | nonsense behaviour |
| 63 | 0.330 | **0.446** | # | driven car to wrong place, can't remember how |
| 57 | 0.316 | **0.357** | # | bad grades in school because of sleepiness |
| 156 | 0.255 | 0.260 | - | work accidents due to sleepiness |

**Item:** Numerical label of this item on the original SDQ

**2A**: Subfactor related to excessive daytime sleepiness

**2B**: Subfactor related to neurological symptoms of narcolepsy

**Source**: This item appeared on a subscale of the original SDQ, as follows:

"**#**": SDQ-NAR subscale

"**-**": Was not a member of an SDQ subscale.

No items co-loaded onto the two subfactors. SDQ items 115, 161, and 156 did not load significantly onto either subfactor, nor on the main factor and so will be omitted.

## Main factor 3: Substances affecting sleep ("SUBST"), α = 0.685

This factor concerns persons who report substance use or abuse and its effect on sleep. Table 6 lists the items that have a loading of 0.4 or higher on this factor, on either the orthogonal or oblique rotations. Factor 3 bears no relation to any subscale of the original SDQ.

We performed a secondary factor analysis on this third factor to determine if it was homogenous. The scree plot (S1 Fig, Part C) shows a clear visual cut-off after the second factor, eigenvalue >1. Subfactor 3A explains 3.5% of the overall dataset variance while subfactor 3B explains 2.1%. Subfactor 3A included all of this factor's items except 3 that are linked to males: gender, item 172 (height), and item 165 (weight at age 20). It seems likely that the last two items are included due to collinearity with the gender item and are not linked to substance use, so they will not be considered legitimate items for this subfactor. Therefore, gender, height, and weight will not be retained, leaving a unitary factor 3.

**Table 6. Item loadings on each subfactor of main factor 3 (orthogonal rotation).**

| Item | 3A | 3B | Source | Brief Item Description |
|---|---|---|---|---|
| 105 | **0.773** | 0.063 | - | use alcohol in order to get to sleep |
| 111 | **0.698** | 0.059 | - | used marijuana to help get to sleep |
| 109 | **0.698** | 0.102 | - | used "street drugs" |
| 107 | **0.697** | 0.100 | - | unaware of actions when drinking |
| 110 | **0.637** | 0.044 | - | used tobacco to help get to sleep |
| 108 | **0.633** | 0.055 | - | tobacco within 2 hours of bedtime |
| 165 | 0.023 | **0.861** | - | weight at age 20 |
| 172 | 0.071 | **0.831** | - | height |
| Gender | -0.158 | **-0.821** | - | 1 = male, 2 = female |

Main Factor 3 relates to subjective sleep difficulties related to substance use or abuse.

**Item:** Numerical label of this item on the original SDQ.

**3A, 3B:** Sub-factorizations of Main Factor 3. Items on 3B *were not kept* for SDQ-2

**Source**: None of these items appeared on any subscale of the original SDQ.

## Main factor 4: Sleep Disordered Breathing ("SDB"), α = 0.762

This factor incorporated 10 of the 12 items from the original SDQ-SA subscale (marked with "@" in Table 7). This table also lists the items that have a loading of 0.4 or higher on this factor, in either orthogonal or oblique rotations.

We performed a secondary factor analysis on Factor 4 to determine if it was homogenous. The scree plot (S1 Fig, Part D) shows two factors with an eigenvalue > 1. Using an orthogonal rotation, subfactor 4A explained 4.1% of the overall dataset variance while subfactor 4B explained 3.6% of the variance. Results for the oblique rotation were not much different. Subfactor 4A contains items pertaining to breathing difficulties such as "I snore." Subfactor 4B contains items concerning gender and weight. Since 4B items also have face validity for an apnea scale, Main Factor 4 will therefore be regarded as unitary. Item 18 ("I suffer from heartburn and choking") did not significantly load on either subfactor nor does it significantly load on the main factor, so does not merit inclusion in SDQ-2.

It is worth noting that items "weight at age 20" and "weight 6 months ago" were originally included in the SDQ in order to investigate the observation by some sleep medicine professionals that apnea patients tend to gain weight over time, often rapidly. Therefore, two new variables "weight gained in the past 6 months" and "weight gained since age 20" were calculated from existing items. The former did not load significantly on any factor, but "weight gain since age 20" loaded onto Factor 4 with weight 0.56. The latter has been drafted as a new item for SDQ-2.

Item 144 ("I have gone through the menopause") will be used only in the female version of the SDQ-2's SDB subscale, which will therefore be one item longer than the male version. Item 172 ("How tall are you?") has been retained on this subscale, since Item 176 (BMI) will not be specifically asked about on the SDQ-2. Rather, a 5 x 5 cell look-up table (see S2 Table) comprised of items 172 (height) and 163 (weight) will be supplied to users to allow them to make an approximate calculation of BMI if this is desired.

## Factor analyses by gender

Gender items were first removed from the dataset, after which it was split into male and female datasets. Separate factor analyses performed on the latter produced similar factor structures and item loadings. Interestingly, subfactors 2A and 2B (narcolepsy and daytime sleepiness)

**Table 7. Item loadings on each subfactor of main factor 4 (orthogonal rotation).**

| Item | 4 | 4A | 4B | Source | Brief Item Description |
|---|---|---|---|---|---|
| 21 | **0.744** | 0.757 | 0.320 | @ | snore loudly and bother others |
| 141 | **0.631** | 0.711 | 0.172 | @ | snoring / breathing problem worse on back |
| 22 | **0.654** | 0.672 | 0.243 | @ | stop breathing in sleep |
| 143 | 0.373 | **0.611** | 0.058 | - | snoring / breathing worse with allergy / infection |
| 23 | **0.479** | 0.572 | 0.111 | @ | awake suddenly gasping for breath |
| 15 | **0.463** | 0.559 | 0.110 | - | sleep disturbs bed partner's sleep |
| 139 | 0.330 | **0.511** | -0.054 | - | nose blocking up when trying to sleep |
| 142 | **0.409** | 0.486 | 0.136 | @ | snoring / breathing problem worse after alcohol |
| 173 | **0.416** | 0.309 | 0.136 | @ | age |
| 71 | **0.390** | 0.283 | 0.250 | @ | high blood pressure *[retained for face validity]* |
| 163 | **0.722** | 0.213 | 0.918 | @ | current weight |
| 165 | **0.504** | 0.018 | 0.794 | - | what was your weight at age 20 |
| 172 | **0.636** | 0.245 | 0.751 | @ | how tall are you |
| 144 | **0.440** | 0.405 | | - | completed menopause? 1 = no. .. 5 = agree strongly |
| "176" | **0.636** | 0.245 | 0.641 | @ | available via "look-up matrix" using 163 & 172 |

**Item:** Numerical label of this item on the original SDQ

**4:** Main Factor 4, prior to factoring into 4A and 4B. However, after consideration of weight questions (see discussion in the text), Factor 4 was retained in its entirety

**4A**: Subfactor related to symptoms of sleep disordered breathing

**4B**: Subfactor related to weight, weight gain, and height

**Source**: If marked "@", item was on the SA (sleep apnea) subscale of the original SDQ

If marked "-", item did not appear on any subscale in the original SDQ.

came out as their own factors in both the male and female datasets, as opposed to being subfactors of a Main Factor in the combined dataset. The other factors in the gender-based analysis did not differ markedly from those in the combined data set.

When the gender-related items were put back into the split analyses, none of the four items for men loaded significantly on any of the main factors. They did load onto their own separate factor which could be considered a subscale of erectile dysfunction, but these items will not be included as a separate subscale on the SDQ-2.

## Discussion

This EFA of the original 176-item SDQ succeeded in its goals. It confirmed and resolved problems with the 4 subscales of the original SDQ, notably their conflation of general daytime sleepiness with neurological symptoms of narcolepsy, and a similar conflation of psychiatric insomnia symptoms with those of simple insomnia disorder. These problems were the result of the limitations of the original discriminant function analysis, so the superiority of EFA on these data is therefore confirmed. In addition, the new factors extracted comprised only 66 items, allowing SDQ-2 to be much quicker to administer than the original questionnaire.

The end result of this analysis is the creation of a new questionnaire, the SDQ-2, whose subscales will consist of the items on the 7 factors described above. These new subscales will be named: *InsPsy* (psychological features of insomnia), *InsSubj* (subjective estimate of insomnia severity), *InsPLM* (insomnia due to limb movements), *EDS* (excessive daytime sleepiness), *NAR* (neurological symptoms of narcolepsy), *SUBST* (sleep effects of substance use), and *SDB* (sleep-disordered breathing). The fact that a similar factor structure emerged when the dataset was divided into male and female respondents suggests that SDQ-2 may evaluate men and

women equally well, although different cut-off scores indicative of the threshold of pathology will likely be needed for each gender, as on the original SDQ. On Main Factor 2, SDQ's conflation of two narcolepsy symptom domains–excessive daytime sleepiness and neurological symptoms of narcolepsy—was resolved by extracting subfactors 2A and 2B. Creation of the EDS subscale also resolved another problem with the original SDQ: it had confounded some of these items within its PLM and PSY subscales. In effect, InsPSY replaces the original PSY subscale, while InsPLM replaces the original PLM scale. The new NAR scale fully replaces the original NAR scale and is now much more specific for the neurological symptoms and signs of narcolepsy, while the new EDS scale independently evaluates sleepiness from whatever cause.

The SDB subscale contained 10 of the 12 original items from the SDQ-SA subscale. While it had two subfactors, breathing difficulties and obesity, having a subscale consisting entirely of weight-related questions was not thought to be useful, so these items were included in Factor 4 to produce a unitary SDB subscale. Although BMI is no longer an item on SDQ-2, its statistical contribution to the SDB scale is preserved by the 5-level height and weight items. For those users wishing to have a numerical estimate of BMI, a look-up matrix using these two items is shown in S2 Table.

An assessment of weight gain on the original SDQ required a clinician to perform calculations using items "current weight" and "weight at age 20". This task will be replaced on SDQ-2 by a new item that asks the patient directly how much weight (if any) they have gained since they were 20 years old; this new item loads on the SDB scale.

Of the five SDQ questions for women, only item 144 "I have gone through the menopause" loaded significantly onto any of the 7 factors—SDB. There is some evidence [41] that lower estrogen levels after menopause cause a lessening of muscle tone, which might aggravate sleep apnea, so this item will be included in the SDQ-2. The other gender-related questions will be discarded.

Being shorter than the original SDQ, the SDQ-2 is expected to have a higher completion rate among patients and to be quicker to complete. As noted above, the 7 new subscales should resolve the original SDQ's inadvertent over-diagnosis of narcolepsy and psychiatric features of insomnia, as identified by Schwegler [8]. The new insomnia subscales also validate the work of Schweer [40] and Brower [37] who proposed insomnia subscales made up of existing SDQ items. In fact, 13 of the 16 items on Sweere's *Insomnia* scale appear on Main Factor 1 in the present study.

Nevertheless, there are some cautionary notes. The sample of SDQ respondents upon which the present study is based was not a random population sample. Most were either referred to sleep disorder clinics with a suspected sleep disorder or were patients with another known disorder that was suspected of causing sleep-wake symptoms, i.e., depression, schizophrenia, alcoholism, chronic fatigue syndrome, or general medical illness. Other respondents were normal university students in psychology courses or healthy control participants in clinical research studies.

In this retrospective dataset, there were no structured interview diagnoses of respondents' sleep disorders, psychiatric illnesses, or general medical conditions. Likewise, no racial / ethnic / cultural information was available. Although some clinical diagnoses unrelated to sleep were available, these were made by numerous clinicians in different hospitals in different years and without uniform diagnostic criteria.

Another caution is that the data upon which the current study is based were collected over a long time span, some being almost 35 years old. During this interval there were changes in clinical nosology and polysomnographic measurement techniques, not to mention cultural and linguistic changes. It is therefore possible that the response set of the respondents was not

homogeneous, introducing further error variance. On the positive side, such issues would, again, be somewhat mitigated by the large sample size.

The present paper does not show any validation of the new subscales, although most of the respondents had polysomnography. In fact, only 78 persons in this study were normal controls who had no sleep complaints but even they were not age-matched to patients and did not have polysomnography. Therefore, the factor structure reported here needs to be confirmed in larger samples, hopefully including normal persons age-matched to polysomnographically-diagnosed patients. In future validation studies, an ROC analysis of each SDQ-2 subscale will be very informative and will likely find different cut-off scores in men versus women to indicate the threshold for pathology.

Previous studies of the original SDQ-SA subscale by other authors—using patients that were not included in the present study—have shown its accurate identification of sleep apnea. Hopefully, future convergent and divergent validity studies of the new SDQ-2 subscales using psychometric and physiological assessments will show even greater precision.

The 7 subscales proposed in the present article will allow re-analysis of SDQ datasets gathered by different investigators over the years; these contain the responses of tens of thousands of persons. Especially in those datasets from controlled studies of PTSD, CFS, depression, and anxiety, it is possible that new insights into those patients' symptoms could be obtained via these new subscales.

It should be noted that all of the subscales extracted by EFA in this article are based on the exact wording and item-numbering of the original SDQ. It is for this reason that retrospective analyses can be done on old SDQ data. However, on the new SDQ-2 questionnaire there will be substantial changes in item-numbering plus a change in the form of Likert response choices: from 1 .. . 5 on the original SDQ to 0 .. . 4 on the SDQ-2. There will also be re-wording of several items to give greater precision in symptom description, plus a general rewording to avoid adjectives about magnitude or intensity in the items' stems. For those users wishing to continue using the original SDQ, it is suggested that the 4 original subscales be retired in favour of the 7 new ones. For those users of the new 66-item SDQ-2, it will only be possible to score it by using the 7 new scales.

Future goals for the SDQ-2 include studies of test-retest reliability plus validity studies using other questionnaires and physiological measures. It is hoped that these will be done both in the original English version and in the translated versions, which in some cases would be the first validation of those translations. Such validation would facilitate comparison studies between occupational, cultural, linguistic, and geographical groups.

A prospective study of 800 members of a large urban police force has been completed, in which the 7 new SDQ-2 subscales were concurrently validated against a number of published sleep and psychological questionnaires: ESS [9], ISI [42], STOP-Bang [15], Berlin Questionnaire [14], GAD-7 [43], PHQ-9 [44], MIES [45], PCL-5 [46], and CAGE [47]. A manuscript on this subject is in preparation.

The new SDQ-2 should have more clinical utility than the original SDQ, in that it identifies a greater range of sleep pathologies and does it with better precision. It may also supplant various shorter single-disorder questionnaires.

The full formatted text of the SDQ-2 is attached to this paper as S1 Questionnaire and a form for scoring all its subscales is attached as S1 File. In addition, the full dataset upon which this paper is based (167 variables on 2131 respondents) is attached as an Excel file in S1 Dataset.

The Royal's Institute of Mental Health Research, affiliated with the University of Ottawa (https://www.theroyal.ca/research/sleep-mental-health-research)) will be hosting a collection

of written materials and scientific data about SDQ-2. The Institute will also host an email address for copyright inquiries and technical questions (SDQ2@theroyal.ca).

## Supporting information

**S1 Fig. Scree plots of the subfactors of all 4 major factors.**
(DOCX)

**S1 Table. Content description of factors 5–19.**
(DOCX)

**S2 Table. Look-up matrix to estimate BMI from height & weight.**
(DOCX)

**S1 Appendix. Collinearity diagnostics of main factors.**
(DOCX)

**S1 Questionnaire. Fully formatted text of SDQ-2.**
(PDF)

**S1 File. Graphics to score SDQ and SDQ-2 subscales from raw data.**
(DOCX)

**S1 Dataset. Raw data (n = 2131, v = 167) in an excel sheet.**
(XLSX)

**S1 Checklist. STROBE statement—checklist of items that should be included in reports of observational studies.**
(DOCX)

## Acknowledgments

Sleep Disorders Clinic, Royal Ottawa Mental Health Center (Drs. Alan Douglass, Elliott Lee, and Louis Soucy) and The University of Ottawa Institute of Mental Health Research at the Royal, Ottawa, Ontario, Canada

University of Michigan Psychiatry Department (Drs. John Greden, Mark Demitrack, Israel Liberzon, Kirk Brower, Alan Douglass) and Neurology Sleep Disorders Clinic (Drs. Michael Aldrich, Ronald Chervin), UM Medical Center, Ann Arbor, Michigan, USA

Sleep Disorders Center, Stanford University Medical Center (Drs. Christian Guilleminault, William C. Dement, and Sharon Keenan), Palo Alto, California, USA.

Sleep Disorders Clinic, University of Alberta Hospital (Drs. Alan Douglass and Godfrey Man; Rhoda Schreiner RPsgT), Edmonton, AB, Canada.

Psychology Dept., University of Alberta (Dr. Don Kuiken), Edmonton, AB, Canada

Harborview Medical Center, University of Washington (Dr. Dedra Buchwald), Seattle, Washington, USA

Sleep Disorders Clinic, Isaac Walton Killam Hospital, Dalhousie University (Dr. Rachel Moorehouse), Halifax, Nova Scotia, Canada.

## Author Contributions

**Conceptualization:** Kathleen Biard, Joseph De Koninck, Alan B. Douglass.

**Data curation:** Kathleen Biard, Alan B. Douglass.

**Formal analysis:** Alan B. Douglass.

**Methodology:** Kathleen Biard, Alan B. Douglass.

**Project administration:** Joseph De Koninck.

**Software:** Kathleen Biard, Alan B. Douglass.

**Supervision:** Joseph De Koninck.

**Writing – original draft:** Kathleen Biard, Joseph De Koninck, Alan B. Douglass.

**Writing – review & editing:** Joseph De Koninck, Alan B. Douglass.

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
