## [Decision Letter · Decision Letter 0]

14 Aug 2023

PONE-D-23-17429Creation of a Shortened Version of the of the Sleep Disorders Questionnaire (SDQ)PLOS ONE

Dear Dr. DOUGLASS,

Thank you for submitting your manuscript to PLOS ONE. After careful consideration, we feel that it has merit but does not fully meet PLOS ONE’s publication criteria as it currently stands. Therefore, we invite you to submit a revised version of the manuscript that addresses the points raised during the review process.

We look forward to receiving your revised manuscript.

Kind regards,

Mohammad Hossein Ebrahimi

Academic Editor

PLOS ONE

Journal Requirements:

"I have read the journal's policy and the authors of this manuscript have the following competing interests: Dr. Douglass retains a copyright on the resulting SDQ-2 questionnaire and could benefit from its commercialization" 

Reviewers' comments:

Reviewer's Responses to Questions

**Comments to the Author**

1. Is the manuscript technically sound, and do the data support the conclusions?

Reviewer #1: Yes

Reviewer #2: Yes

2. Has the statistical analysis been performed appropriately and rigorously? 

Reviewer #1: Yes

Reviewer #2: Yes

3. Have the authors made all data underlying the findings in their manuscript fully available?

Reviewer #1: Yes

Reviewer #2: Yes

4. Is the manuscript presented in an intelligible fashion and written in standard English?

Reviewer #1: Yes

Reviewer #2: Yes

5. Review Comments to the Author

Reviewer #1: SDQ is not translated into my native language. One of the reasons for this is that in clinical situations, it takes much time to answer all the 176 questions. In outpatient clinics and sleep centers, shorter questionnaires such as ESS, PSQI, AIS and SDS (Self-rating Depression Scale) are preferred and acquired.

This research sheds light on this weak point of the original version of SDQ, making the new shortened SDQ-2.

It might be better if the whole set of questions in SDQ-2 is shown in this paper.

I'm looking forward to the validation study of SDQ-2.

Reviewer #2: Summary

The authors used secondary analyses of retrospective anonymized medical record data (N=2131) from multiple sites to conduct exploratory factor analysis (EFA) of the widely used 176-item Sleep Disorders Questionnaire—a self-report measure for identifying sleep disorders that was developed using discriminant function analysis rather than full factor analysis—which has allowed the authors to propose a revised version of the original questionnaire entitled SDQ-2 that reduces the number of items to 66 while identifying new subscales with potentially improved psychometrics.

Strengths

The introduction provides a persuasive rationale for conducting a re-analysis of the SDQ, including a detailed description of the problems that have emerged with use of the original measure (e.g., symptom overlap and significant intercorrelation between subscales, lack of validity for cut-off scores, possibility of unidentified subscales).

The methods are fully outlined and data are available without restriction. Results are described in detail, including analyses of possible gender differences in factor loadings, and tables and figures are clearly labeled and help to illustrate results without being redundant.

Limitations are appropriately identified (protracted data collection time frame, non-random sample, no racial/ethnic demographic information, no corroborating clinician data). Relevant and interesting future studies are suggested (validation of new SDQ-2, re-analysis of original SDQ datasets).

The writing is clear and accessible even to those without specialized knowledge of sleep medicine. References are adequate and appropriate.

Significance

The newly developed 66-item SDQ-2 appears to offer a comprehensive but concise assessment of possible sleep disorders to support diagnosis and referral, especially in non-specialized medical settings such as primary care.

6. PLOS authors have the option to publish the peer review history of their article (what does this mean?). If published, this will include your full peer review and any attached files.

Reviewer #1: No

Reviewer #2: No

---

## [Author Response · Author response to Decision Letter 0]

18 Sep 2023

Response to reviewers is in the file "Rebuttal Letter.docx", which was uploaded today.

---

## [Decision Letter · Decision Letter 1]

15 Jan 2024

Creation of a Shortened Version of the Sleep Disorders Questionnaire (SDQ)

PONE-D-23-17429R1

Dear Dr. DOUGLASS,

We’re pleased to inform you that your manuscript has been judged scientifically suitable for publication and will be formally accepted for publication once it meets all outstanding technical requirements.

Kind regards,

Mohammad Hossein Ebrahimi

Academic Editor

PLOS ONE

Additional Editor Comments (optional):

Reviewers' comments:

Reviewer's Responses to Questions

**Comments to the Author**

1. If the authors have adequately addressed your comments raised in a previous round of review and you feel that this manuscript is now acceptable for publication, you may indicate that here to bypass the “Comments to the Author” section, enter your conflict of interest statement in the “Confidential to Editor” section, and submit your "Accept" recommendation.

Reviewer #3: All comments have been addressed

2. Is the manuscript technically sound, and do the data support the conclusions?

Reviewer #3: Yes

3. Has the statistical analysis been performed appropriately and rigorously? 

Reviewer #3: Yes

4. Have the authors made all data underlying the findings in their manuscript fully available?

Reviewer #3: Yes

5. Is the manuscript presented in an intelligible fashion and written in standard English?

Reviewer #3: Yes

6. Review Comments to the Author

Reviewer #3: This study presents several novel contributions to the field of sleep disorder diagnosis, especially in terms of reassessing and refining a pre-existing diagnostic tool. The application of exploratory factor analysis to the SDQ and the creation of the SDQ-2, with its reduced and more focused set of items, represent significant advancements.

7. PLOS authors have the option to publish the peer review history of their article (what does this mean?). If published, this will include your full peer review and any attached files.

Reviewer #3: **Yes: **Muayad Albadrani

---

## [Editor Report · Acceptance letter]

26 Jan 2024

PONE-D-23-17429R1 

PLOS ONE

Dear Dr. Douglass, 

I'm pleased to inform you that your manuscript has been deemed suitable for publication in PLOS ONE. Congratulations! Your manuscript is now being handed over to our production team.

Kind regards, 

on behalf of

Dr. Mohammad Hossein Ebrahimi 

Academic Editor

PLOS ONE